# Recombinant Adeno-Associated Viral Vector-Mediated Gene Transfer of *hTBX18* Generates Pacemaker Cells from Ventricular Cardiomyocytes

**DOI:** 10.3390/ijms23169230

**Published:** 2022-08-17

**Authors:** Melad Farraha, Renuka Rao, Sindhu Igoor, Thi Y. L. Le, Michael A. Barry, Christopher Davey, Cindy Kok, James J.H. Chong, Eddy Kizana

**Affiliations:** 1Sydney Medical School, the University of Sydney, Sydney 2006, Australia; 2Centre for Heart Research, the Westmead Institute for Medical Research, Sydney 2145, Australia; 3Department of Cardiology, Westmead Hospital, Sydney 2145, Australia; 4School of Physics, the University of Sydney, Sydney 2006, Australia

**Keywords:** adeno-associated viral vector, cardiomyocytes, gene therapy, heart, hTBX18, pacemaker cells

## Abstract

Sinoatrial node dysfunction can manifest as bradycardia, leading to symptoms of syncope and sudden cardiac death. Electronic pacemakers are the current standard of care but are limited due to a lack of biological chronotropic control, cost of revision surgeries, and risk of lead- and device-related complications. We therefore aimed to develop a biological alternative to electronic devices by using a clinically relevant gene therapy vector to demonstrate conversion of cardiomyocytes into sinoatrial node-like cells in an in vitro context. Neonatal rat ventricular myocytes were transduced with recombinant adeno-associated virus vector 6 encoding either *hTBX18* or green fluorescent protein and maintained for 3 weeks. At the endpoint, qPCR, Western blot analysis and immunocytochemistry were used to assess for reprogramming into pacemaker cells. Cell morphology and Arclight action potentials were imaged via confocal microscopy. Compared to *GFP*, *hTBX18*-transduced cells showed that *hTBX18*, *HCN4* and *Cx45* were upregulated. *Cx43* was significantly downregulated, while sarcomeric α-actinin remained unchanged. Cardiomyocytes transduced with *hTBX18* acquired the tapering morphology of native pacemaker cells, as compared to the block-like, striated appearance of ventricular cardiomyocytes. Analysis of the action potentials showed phase 4 depolarization and a significant decrease in the APD50 of the *hTBX18*-transduced cells. We have demonstrated that rAAV-*hTBX18* gene transfer to ventricular myocytes results in morphological, molecular, physiological, and functional changes, recapitulating the pacemaker phenotype in an in vitro setting. The generation of these induced pacemaker-like cells using a clinically relevant vector opens new prospects for biological pacemaker development.

## 1. Introduction

The sinoatrial node (SAN) is a group of highly specialized cells, containing less than 10,000 genuine pacemaker cells that maintain the regular beating of the mammalian heart [1,2]. The SAN, however, can become defective via several means, including myocardial infarction, cardiomyopathy, and genetic defects but most prevalently because of ageing [3,4]. This condition affects approximately 1 in 600 cardiac patients older than 65 years and accounts for 50 percent or more of permanent pacemaker insertions in the United States alone [4,5,6], with the incidence rising due to the world’s ageing population [7]. Dysfunction of the SAN leads to cardiac rate control issues including bradycardia, with severe cases resulting in sudden cardiac death [8,9]. To date, there are no known cures for SAN dysfunction, with the only viable management option being the implantation of an electronic pacemaker. Implantable electronic pacemaker technology has continued to evolve since its development six decades ago [10,11]. Although effective, these devices and their subsequent surgeries present their own issues ranging from insertion complications [11,12,13], follow-up surgeries required to replace batteries and to correct device problems, but most seriously, infections that require the removal of the device for effective antibiosis to be administered [11,14]. The complications associated with electronic pacemakers and the existence of a need for better treatment of SAN dysfunction have motivated research into discovering more effective and innovative treatment options, including gene transfer of ion channels and developmental transcription factors necessary for SAN formation [14].

Over the past two decades, complex studies have been carried out to gain a greater understanding of the pathways that drive the formation, migration, and assembly of cardiac tissue, the pacemaker and the conduction system [15]. Early morphological studies of the SAN described 3 cell types: nodal, intermediate, and elongated cells, which are thought to drive the pacemaker [16]. The developmental biology of these cells is now known to arise from the sinus venosus, which is heavily regulated by numerous transcription factors as it develops [17]. The transcription-regulating candidates for embryonic SAN development include Shox2, TBX3, TBX5 and TBX18 [17]. Shox2 is a negative regulator of Nkx2.5 (an early precardiac marker essential for the development of the cardiac conduction system [18,19,20,21]) in the sinus venosus, and Shox2-deficient mouse and zebrafish embryos displayed bradycardia [18,22]. TBX3 is a potent regulator of SAN specialization, with developmental errors resulting from either deficiency or ectopic expression [23]. TBX5, which shows an inverse correlation between its dosage and abnormal cardiac morphogenesis, is a positive regulator of Shox2 and TBX3 [24]. However, upstream of all these factors is TBX18. Mesenchymal progenitor cells expressing TBX18 define the sinus venosus and differentiate de novo into SAN cells. TBX18 is required for the differentiation of ventricular cardiomyocytes, triggering the genetic pathways related to SAN development, becoming undetectable by birth [25,26]. Therefore, understanding the importance of TBX18 in this context has informed the direction in which biological pacemaker research has evolved.

The first attempt at using recombinant human T-box 18 (*hTBX18*) gene therapy to develop a biological pacemaker involved its overexpression via an adenoviral vector in ventricular cardiomyocytes of adult guinea pigs using a bradycardic disease model [1]. Transduction of *hTBX18* induced conversion of ventricular cardiomyocytes into SAN-like cells, which resembled the endogenous SAN. As opposed to previous approaches, no single determinant of excitability was selectively overexpressed. Rather, the entire gene expression profile was altered, resulting in fundamental phenotypic changes to the cells. Furthermore, the in vivo reprogramming by *hTBX18* created a pacemaker rhythm in the guinea pig hearts that corrected the disease model.

Following on from the small animal model, *hTBX18* was again delivered using an adenoviral vector via a minimally invasive technique in a pig model of complete heart block [27]. The vector encoding *hTBX18* was delivered to the His-bundle region by an endocardial needle catheter. The results showed that animals transduced with *hTBX18* had stable pacemaker activity originating from the injection site while also having a higher heart rate, more diurnal heart rate variation, decreased reliance on the backup electronic pacemaker and an increased physiologic autonomic response. Moreover, there were no signs of pro-arrhythmia or systemic adverse effects. While these results were the most promising to date, biological pacing in this model peaked at day 7–8 and waned by day 14. The decline in pacing was due to the predictable host immune response to the adenoviral vector employed [1,28,29,30,31].

Newer attempts at using *hTBX18* have attempted to use both gene- and gene-cell hybrid-based approaches to generate SAN-like pacemaker cells, both in vivo or in vitro, before implantation [32,33,34,35,36,37]. Further studies are required to assess the engraftment potential, pacing efficacy and any proarrhythmic effects of these strategies [32].

There are still many inherent limitations associated with the development of gene therapy. The use of viral vectors to deliver the necessary genes presents its own problems. It is appreciated that adenoviral vectors can only provide transient expression for pacemaker function largely due to the strong and rapid inflammatory and immune responses triggered by the vector, resulting in vector clearance [1,28,29,30,31]. Other vectors, such as lentivirus vectors, are not cardiotropic and carry an attendant risk of insertional mutagenesis [38,39]. Therefore, the choice of viral vector will predictably impact the duration of expression and subsequent functional effects of the transgene. Recombinant adeno-associated viral vectors (rAAV) present a promising alternative for several reasons. A subset of natural capsid subtypes are highly cardiotropic, minimally immunogenic compared to other vectors, allow for longer-term gene expression, and have been used in several human clinical trials highlighting their efficacy as a gene therapy vector, and further research is being conducted via capsid engineering to create rAAVs with enhanced cardiotropism and the ability to evade the immune response [40,41,42,43].

Therefore, the aim of this study was to design and develop an rAAV construct containing *hTBX18* and to validate its functionality in reprogramming ventricular myocytes into SAN-like pacemaker cells in an in vitro context.

## 2. Results

### 2.1. rAAV Vectors Expressing GFP or hTBX18 Showed Highly Efficient Transduction in NRVMs

Neonatal rat ventricular myocytes (NRVM) transduced for 3 weeks with rAAV6-*GFP* (Figure 1A) and rAAV6-*hTBX18* (Figure 1B) at an MOI of 30,000 showed transduction efficiencies greater than 90%, as compared to the non-transduced controls, when visualized via microscopy and analyzed via flow cytometry (Figure 1C).

### 2.2. NRVMs Transduced with rAAV6-hTBX18 Faithfully Reproduced the Distinctive Molecular and Physiological Characteristics of Pacemaker Cells

Three weeks following transduction with rAAV6-*GFP* (control) or rAAV6-*hTBX18*, the following 5 genes associated with pacemaker cells were examined at a mRNA, protein and immunocytochemical level: *hTBX18*, hyperpolarization-activated cyclic nucleotide-gated channel 4 (*HCN4*), gap junction connexin 43 (*Cx43*), gap junction connexin 45 (*Cx45*) and sarcomeric α-actinin (*SαA*).

#### 2.2.1. Analysis of mRNA Expression Showed Changes Characteristic of Pacemaker Cells

Following transduction with rAAV6-*GFP* or rAAV6-*hTBX18* (n = 8), expression levels of the different genes were analyzed by qPCR, normalized to GAPDH expression and the two groups were compared. The following results were observed: *hTBX18* expression in the rAAV6-*hTBX18*-transduced cells was 1145 times higher as compared to the rAAV6-*GFP*-transduced cells (Figure 2A). *HCN4* in the rAAV6-*hTBX18*-transduced cells was 3 times higher as compared to the rAAV6-*GFP*-transduced cells (Figure 2B). *Cx43* expression in the rAAV6-*hTBX18*-transduced cells was 3.5 times lower as compared to the rAAV6-*GFP*-transduced cells (Figure 2C). *Cx45* expression in the rAAV6-*hTBX18*-transduced cells was 1.6 times higher as compared to the rAAV6-*GFP*-transduced cells but did not achieve statistical significance (Figure 2D). *SαA* expression in the rAAV6-*hTBX18*-transduced cells was equal to that of the rAAV6-*GFP*-transduced cells (Figure 2E).

#### 2.2.2. Analysis of Protein Expression Showed Changes Characteristic of Pacemaker Cells

Following transduction with rAAV6-*GFP* or rAAV6-*hTBX18* (n = 7), the changes in expression levels of the following proteins relative to *β-actin* were observed (Figure 3A): *hTBX18* in the rAAV6-*hTBX18*-transduced cells was higher as compared to the rAAV6-*GFP*-transduced cells (Figure 3B). *HCN4* in the rAAV6-*hTBX18*-transduced cells was also higher as compared to the rAAV6-*GFP*-transduced cells (Figure 3C). *Cx43* in the rAAV6-*hTBX18*-transduced cells was lower as compared to the rAAV6-*GFP*-transduced cells (Figure 3D). *Cx45* in the rAAV6-*hTBX18*-transduced cells was higher as compared to the rAAV6-*GFP*-transduced cells but did not achieve statistical significance (Figure 3E). *SαA* in the rAAV6-*hTBX18*-transduced cells was higher as compared to that of the rAAV6-*GFP*-transduced cells but did not achieve statistical significance (Figure 3F).

#### 2.2.3. Immunocytochemical Analysis of Cells Transduced with rAAV6-*hTBX18* Displayed Distinctive Characteristics of Pacemaker Cells

Following transduction with rAAV6-*GFP* or rAAV6-*hTBX18*, the changes in expression levels of the following markers were observed: *hTBX18* was evident in the nuclei of rAAV6-*hTBX18*-transduced cells only (Figure 4A). *HCN4* showed strong expression in rAAV6-*hTBX18*-transduced cells, which was much more evident as compared to the rAAV6-*GFP*-transduced cells (Figure 4B). *Cx43* showed strong expression with junctional localization in the rAAV6-*GFP*-transduced cells, which was not evident in the rAAV6-*hTBX18*-transduced cells (Figure 4C). *Cx45* showed similar expression patterns in the rAAV6-*hTBX18* and rAAV6-*GFP*-transduced cells (Figure 4D). Additionally, *SαA* expression in the rAAV6-*hTBX18* transduced cells showed disorganized myofibril structures as compared to that of the rAAV6-*GFP*-transduced cells, which showed well-organized and striated myofibril structures (Figure 4E).

### 2.3. NRVMs Transduced with rAAV6-hTBX18 Faithfully Recapitulate the Morphological Features of Pacemaker Cells

In addition to the molecular and physiological changes, rAAV6-*hTBX18*-transduced cells underwent distinct morphological changes. *hTBX18* cells were co-transduced with rAAV6-*GFP* to visualize under fluorescent microscopy. NRVMs originally showed large, thick, block-like structures, shown in Figure 5A. Following transduction with rAAV6-*hTBX18*, NRVMs underwent a morphological change and became thin, spindle-like and tapering in structure, similar to native pacemaker cells, as shown in Figure 5B. The mean diameter of the NRVMs (29.38 ± 1.439 µm) was 3 times larger than the rAAV6-*hTBX18*-transduced cells (10.62 ± 0.8472 µm) (Figure 5C). The total area of the NRVMs (5253 ± 453.1 µm) was 5 times larger than the rAAV6-*hTBX18*-transduced cells (1452 ± 132.3 µm) (Figure 5D).

### 2.4. NRVMs Co-Transduced with rAAV6-hTBX18 and LV.Arclight Generated Pacemaker-like Action Potentials

Co-transduction of NRVMs with *LV.Arclight* allowed for electrophysiological characterization of the cells based on a potentiometric fluorescent *GFP* signal (Figure 6), assessed three weeks after transduction. The recordings were made from beating areas of cells in the rAAV6-*GFP* and rAAV6-*hTBX18* cultures. Based on the parameters of the morphology of the action potential (AP) and AP duration at 50% repolarization (APD50), we could distinctly identify three types of AP morphologies. Ventricular myocytes (Figure 6A) and atrial myocytes (Figure 6B) displayed their corresponding action potentials, while *hTBX18*-transduced cells (Figure 6C) displayed a distinctive phase 4 depolarization. Additionally, NRVMs (0.925 ± 0.022 s) had a significantly longer APD50 as compared to rAAV6-*hTBX18*-transduced cells (0.170 ± 0.011 s) (Figure 6D).

## 3. Discussion

In this study, we successfully validated the rAAV vector-mediated gene transfer of *hTBX18* to demonstrate the conversion of NRVMs into SAN-like cells, which recapitulated the biological pacemaker phenotype in vitro.

Previous work had identified that SAN development is tightly regulated by the transcription factors Shox2, TBX3, TBX5 and TBX18 [17]. Using these transcription factors, studies focused on overexpression to either increase the efficiency of the differentiation process of stem cells—either embryonic or induced-pluripotent-derived—into the pacemaker-like phenotype [33,34,35,36,37,44,45,46] or used a viral vector-mediated approach to demonstrate the conversion of cardiomyocytes into pacemaker-like cells [1,27,32,33]. Although all were successful to some degree, TBX18 was shown to be upstream of all these factors [25,26] and demonstrated the greatest potential in achieving functional pacemaker cells in either a stem cell or gene therapy context. However, to address the issues raised with the use of stem cell therapies [47] or gene therapy-mediated approaches using lentiviral or adenoviral vectors [27,32,48,49], we designed and validated the use of an rAAV vector expressing *hTBX18* to reprogram ventricular cardiomyocytes into SAN-like pacemaker cells, which has not been demonstrated previously. Complementing previous findings, our results recognized that the single *hTBX18* transcription factor sufficed for direct conversion of NRVMs into pacemaker-like cells, with the focus on establishing the reliability of conversion using molecular, physiological, morphological, and functional characterization.

rAAV has become the most promising vector for gene therapy clinical trials, including those targeting the heart [50,51,52]. It is derived from a non-pathogenic parental virus, and packaging with capsid subtypes 6, 8 and 9 generate vectors that are cardiotropic and capable of efficient cardiac transduction. This vector generates minimal immunogenicity and can therefore confer long-term gene expression in non-replicating cells such as cardiomyocytes [40,42,43]. It has been successfully used in several human clinical trials, and efforts are continuing to improve its efficiency at transducing target cells and enhancing its ability to evade pre-existing neutralizing antibodies via directed evolution [53,54]. All these characteristics allow for the advancement of biological pacemaker development, addressing the translational issues which have arisen in previous studies.

Since rAAV vectors had not been utilized in this context, we designed a rAAV6 vector expressing *hTBX18*. This vector was confirmed to be functional and highly efficient (>90% transduction efficiency) at expressing *hTBX18* in NRVMs. We chose to use the rAAV6 capsid as it showed the greatest transduction efficiency when compared to capsids considered to be most cardiotropic in rodent models [40,42,55]. This vector was then used to transduce NRVMs for three weeks, and we subsequently analyzed the modification they underwent as they became pacemaker-like cells. Molecular analysis was used to observe changes in five distinct markers: *hTBX18, HCN4, Cx43, Cx45* and *SαA*. These markers were chosen because they allowed us to differentiate between ventricular cardiomyocytes and pacemaker cells.

Adult mammalian ventricular cardiomyocytes do not inherently express the transcription factor *TBX18,* with silencing occurring upon cardiac maturation around the time of birth [1,26]. *hTBX18* expression was significantly upregulated at an mRNA and protein level because of transduction with rAAV6-*hTBX18.* Immunocytochemistry confirmed the presence of *hTBX18* in rAAV6-*hTBX18*-transduced cells only and was shown to be localized to the nucleus. With the knowledge that *hTBX18* was being overexpressed, the other markers were similarly examined to confirm they changed in a manner reflective of pacemaker cells.

Hyperpolarization-activated cyclic nucleotide-gated channels (HCN) produce the I_f_ current, which is responsible for diastolic depolarization in the SAN cells [56]. *HCN1*, *HCN2* and *HCN4* are the main subtypes of the HCN channels in the heart [57], with the SAN cells containing the highest proportion of *HCN4*, which is important in phase 4 automatic depolarization of pacemaker cells [58,59,60]. *HCN4* expression was significantly upregulated in the rAAV6-*hTBX18* transduced cells. The underlying mechanism responsible for this upregulation is due to the *HCN4* promoter being activated by *hTBX18* [1], which directly increases the expression of *HCN4* and enhances the conversion of the cardiomyocytes into a pacemaker-like phenotype.

The gap junction proteins (connexins) in cardiomyocytes permit intercellular coupling and conduction of electrical signals to allow for synchronized contraction of the heart. They also specifically shield the SAN from the hyperpolarizing environment of the atrial tissue [61,62]. There are three main connexins that are expressed in the heart: *Cx40, Cx43 and Cx45* [62,63,64]. *Cx45* predominates in the SAN to slow conduction, while *Cx43* is found in all other chambers except the conduction system to maintain conduction integrity [64,65]. To distinguish the difference between ventricular cardiomyocytes and pacemaker-like cells, *Cx43* and *Cx45* expression levels were analyzed as they differed the most with changes in expression levels [65,66]. As predicted, *Cx43* expression was significantly downregulated in the rAAV6-*hTBX18*-transduced cells. *Cx45* expression was upregulated in the rAAV6-*hTBX18*-transduced cells, although this did not achieve statistical significance. These findings in connexin expression concur with previous literature that summarizes *Cx43* to be significantly downregulated in pacemaker cells, while *Cx45* either remains the same or increases as compared to ventricular myocytes [1,64,66]. Once again, the underlying mechanism responsible for this significant downregulation of *Cx43* involves *hTBX18* selectively repressing the *Cx43* promoter over other connexins, facilitating the slow and safe propagation of action potentials within the SAN before disseminating to the working myocardium [66].

Cardiac *SαA* functions to anchor myofibrillar actin thin filaments to Z-discs within the muscle fibers of cardiac cells, giving rise to their structure and organized striations [67]. Its expression has been previously described to be lower and disorganized in the SAN as compared to the adjacent right atrium [1]. *SαA* expression was relatively unchanged in rAAV6-*hTBX18*-transduced cells. However, when analyzed via immunocytochemistry, it was evident that there was myofibrillar disorganization and a breakdown in the structure of *SαA* (Figure 4E). This is most likely attributed to the change in morphology the cells undergo as they convert into pacemaker cells, requiring the myofibrillar structures to break down and re-arrange as the cells move from thick, block like structures into thin, spindle shaped ones.

In addition to the molecular and physiological changes which occurred, changes in the morphology of the cellular structure were also observed, further validating the conversion of the ventricular cardiomyocytes into pacemaker-like cells. Native SAN pacemaker cells present a distinct morphology: they are smaller [68] and exhibit less-organized myofibrils [2,69] than ventricular cardiomyocytes. rAAV6-*hTBX18*-transduced cells underwent distinct morphological changes, starting as thick, block-like structures before becoming thin, spindle-like and tapering in morphology, indicative of pacemaker cells (Figure 5). The converted cells were, on average, 5 times smaller and 3 times thinner, resembling SAN cells [64,69].

The final measure of successful conversion to pacemaker-like cells involved analysis of the action potentials generated by rAAV6-*hTBX18* cells. Action potentials were recorded using the Arclight construct—a variant of the Ciona intestinalis voltage-sensitive-based fluorescent protein voltage sensor [70,71,72,73]. This approach was able to successfully differentiate between atrial, ventricular and pacemaker-like action potentials based on the action potential morphology. Atrial and ventricular myocytes displayed their distinct action potentials, while rAAV6-*hTBX18*-transduced cells displayed abbreviated, atrial-like potentials with spontaneous phase 4 depolarization, the hallmark of pacemaker cells. There was a significant decrease in the APD50 of rAAV6-*hTBX18*-transduced cells. Taken together, these changes in the action potential of rAAV6-*hTBX18*-transduced cells are consistent with an electrophysiological phenotypic change from ventricular to pacemaker cells. Our findings are consistent with others found in literature [1,33,71,73,74].

There are two main limitations to this approach for creating pacemaker cells using rAAV. As previously outlined, *hTBX18* is an embryonic transcription factor that ceases to be expressed following complete SAN development and birth. As such, the effects of persistent expression of *hTBX18* using rAAV are unknown beyond the study period of 3 weeks. Further investigations need to be undertaken to determine if there are any detrimental effects following long-term expression of *hTBX18* both in vitro and in vivo. This limitation could be addressed by the use of a tissue-specific promotor with an inherent capacity for regulation or expression cassette regulatory elements designed to reduce transgene expression following somatic reprogramming. The other main limitation is that the in vivo potential of an AAV-based vector to induce reprogramming is unknown. The vector copy numbers of functional rAAV particles injected in vivo will be lower as compared to adenovirus-based vectors due to the much higher titers achieved with the latter. Therefore, the threshold for somatic reprogramming and phenotypic conversion may be more challenging to achieve with rAAV compared to established protocols with adenoviral vectors. This limitation could be addressed by employing rAAV with capsids engineered to be more cardiotropic.

## 4. Materials and Methods

### 4.1. Molecular Cloning of hTBX18 Gene into a rAAV Expression Vector

The human TBX18 gene without the C-terminal Flag–ZsGreen tag was PCR amplified from pLV-TBX18-Flag–ZsGreen [66] using custom-designed *hTBX18* forward and reverse primers. The forward primer added a BamHI restriction site and Kozak consensus sequence (5′-ATGGATCCACCACCATGGCCGAGAAG-3′). The reverse primer added a stop codon followed by a SpeI restriction site (5′-CGACTAGTTCAGACCATATGTGCAG-3′). Digestion using BamHI and SpeI sites allowed *hTBX18* to be subcloned into an AAV_CBA_WPRE_Empty expression vector containing the desired chicken beta actin (CBA) promoter and woodchuck post-transcriptional regulatory element (WPRE) (Figure 7). This generated our desired construct: AAV_CBA_*hTBX18*_WPRE (7.2 kB). The control construct, AAV_CBA_*GFP*_WPRE (6.2 kB), encoded for *GFP*. Generated constructs were verified to encode the correct target gene by Sanger DNA sequencing (AGRF, Sydney, Australia).

### 4.2. rAAV6 Production and Purification

rAAV6 vectors were prepared with a two-plasmid system as previously described with modification [43,75,76,77]. Briefly, HEK293T cells were seeded in Falcon 150 mm tissue culture dishes (Corning Life Sciences, Teterboro, NJ, USA) in DMEM (Lonza, Basel, Switzerland) and 10% fetal bovine serum (FBS) (Sigma-Aldrich, St. Louis, MO, USA) media. Cells were transfected with plasmid DNA 24 h later using 22.5 µg of the rAAV expression vector (*GFP* or *hTBX18*) and 45µg of the packaging/helper plasmid pDGM6 [77] per plate by calcium phosphate precipitation. Sixteen hours later, the media was changed to fresh DMEM and 10% FBS. Forty-eight hours following transfection, clarified media from rAAV 2/6-transfected cells were collected and stored at 4 °C until purification.

Vector was purified from the AAV 2/6 supernatant as previously described [43,75,76,78]. In brief, the supernatant was incubated with 40% polyethylene glycol (Sigma-Aldrich, St. Louis, MO, USA) for >3 h on ice. The precipitate was spun down and gently resuspended in 10 mM Tris, 10 mM Bis Tris Propane (Sigma-Aldrich, St. Louis, MO, USA) before benzonase (Sigma-Aldrich, St. Louis, MO, USA) treatment for 30 min at 37 °C. The supernatant was then layered on a 12.5 mL cesium chloride (CsCl) (Affymetrix, Santa Clara, CA, USA) density gradient (7 mL of 1.37 g/mL CsCl layered on 1 mL of 1.5 g/mL CsCl) and ultracentrifuged at 36,000 rpm for 40 h at 16 °C in an Optima XPN-100 ultracentrifuge (Beckman Coulter, Brea, CA, USA).

Following ultracentrifugation, 1 mL fractions were collected, and real-time quantitative PCR (qPCR) was performed directly on aliquots from the fractions. Fractions with the highest concentration of vector particles were pooled and dialyzed for three rounds against 1 × phosphate-buffered saline (PBS) (Thermo Fisher Scientific, Waltham, MA, USA) containing 5% glycerol (Sigma-Aldrich, St. Louis, MO, USA), before concentration using vivaspin 300,000 MWCO ultrafiltration tubes (Sartorius, Göttingen, Germany). A final qPCR was performed to determine the titer of the concentrated vector stocks. The final qPCR titer was ~2 × 10^12^ vector genomes (vg)/mL.

### 4.3. Neonatal Rat Ventricular Myocyte Isolation, Culture and rAAV Transduction

Primary cultures of neonatal rat ventricular myocytes (NRVM) were prepared by enzymatic digestion of ventricles obtained from neonatal (2–3 day old) Sprague-Dawley rats, as previously described [79,80,81]. Briefly, atria and major vessels were trimmed from the ventricles before being incubated overnight at 4 °C in Hank’s balanced salt solution (HBSS, Lonza, Basel, Switzerland) containing 0.1% trypsin (Affymetrix, Santa Clara, CA, USA). The following morning, the hearts were dissociated in 4 to 5 rounds of incubation with 0.1% collagenase solution (Worthington Biochemical Corporation, Lakewood, NJ, USA). Isolated cells were enriched for myocytes by pre-plating cell suspensions in T150 flasks (Corning Life Sciences, New York, NY, USA) for 1–2 h. Fibroblasts attach much more rapidly than myocytes, with pre-plating removing most of the fibroblasts from the cardiac cell suspension. A cell count was performed on the enriched suspension, after which cells were plated in gelatin-coated 6- or 24-well tissue culture plates (Corning Life Sciences, New York, NY, USA) or glass-bottom confocal dishes (Nunc, Roskilde, Denmark) at a density of 1.1 × 10^5^ cells/cm^2^. The cells were plated in culture media (M199 media Sigma-Aldrich, St. Louis, MO, USA) supplemented with 0.35 µg/mL glucose (Sigma-Aldrich, St. Louis, MO, USA), 2 mM L-glutamine (Sigma-Aldrich, St. Louis, MO, USA), 1 × HEPES buffer (Sigma-Aldrich, St. Louis, MO, USA), 1 × MEM non-essential amino acids (Sigma-Aldrich, St. Louis, MO, USA), 0.2 ng/mL vitamin B_12_ (Sigma-Aldrich, St. Louis, MO, USA), 5 U/mL penicillin (Sigma-Aldrich, St. Louis, MO, USA) and 10% fetal bovine serum (FBS, Sigma-Aldrich, St. Louis, MO, USA)). The following morning, the cells were washed with two rounds of PBS involving vigorous shaking to dislodge the dead cells, then replenished with fresh media. Every second day, the media was changed to maintenance media (M199 + 2% FBS (including supplements)) for maintenance of the cells.

Following the switch to maintenance media, cells were either left non-transduced or transduced with rAAV6-*GFP* or rAAV6-*hTBX18* at a pre-determined (data not shown) MOI of 30,000 to initiate viral vector transduction.

### 4.4. Messenger RNA Extraction and Complementary DNA Synthesis

Three weeks post-transduction with rAAV, NRVMs were trypsinized (0.1% trypsin and 0.04% EDTA (Sigma-Aldrich, St. Louis, MO, USA)), washed once with PBS and pelleted before messenger RNA (mRNA) was extracted using an mRNA isolation kit according to the manufacturer’s instructions (Qiagen, Venlo, The Netherlands). Once eluted, mRNA was stored in aliquots at −80 °C.

The mRNA samples were converted to first strand cDNA using the M-MLV RT DNA polymerase kit according to the manufacturer’s instructions (Promega, Madison, WI, USA). The appropriate no-RNA and no-enzyme controls were used to confirm cDNA integrity. cDNA was diluted to 5 ng/µL for use with real-time polymerase chain reaction (RT-PCR) analysis.

### 4.5. Real-Time PCR Analysis

RT-PCR was used to evaluate the changes in gene expression between the different NRVM treatment groups. In brief, primers (Table 1) were designed and validated for the different genes to be examined. Reactions were prepared in a 384-well plate (Biorad, Hercules, CA, USA) and performed in 10 µL volumes using the following mix: 1 × Sensifast SYBR RT-PCR master mix (Bioline, Memphis, TN, USA), 0.4 µM primers, 1 µL of template cDNA and molecular biology grade sterile water (Lonza, Basel, Switzerland). The plates were run in a Biorad CFX384 PCR thermocycler (Biorad, Hercules, CA, USA) using the following conditions: denaturation at 95 °C for 2 min, followed by 40 cycles of denaturing for 5 s at 95 °C, annealing for 10 s at 60 °C and extension for 15 s at 72 °C, ending with a melt cycle. Melt curves and CT values for GAPDH across all samples were confirmed to be consistent before analysis. The cycle threshold values obtained from duplicate samples were analyzed for relative expression changes.

### 4.6. Protein Extraction and Immunoblotting

Three weeks following transduction with rAAV6, NRVMs in 6-well plates were incubated for 5 min with 200 µL of ice-cold lysis solution containing RIPA buffer (Sigma-Aldrich, St. Louis, MO, USA) and 1% (*v*/*v*) protease inhibitor cocktail (Sigma-Aldrich, St. Louis, MO, USA). Lysed samples were pelleted, and protein supernatant was collected. Protein estimation was performed using the Pierce BCA Protein Assay Kit (Thermo Scientific, Waltham, MA, USA), and the absorbance at ~562 nm was measured using a Victor Plate Reader (PerkinElmer Life Sciences, Waltham, MA, USA) alongside a standard curve, to infer the concentration of the protein samples.

Once quantified, equal amounts of protein were diluted in 4× Laemmli sample buffer (Bio-Rad, Hercules, CA, USA), denatured at 95 °C for 5 min and loaded onto 10% sodium dodecyl sulphate-polyacrylamide gel electrophoresis (SDS-page) gels. Gels were electrophoresed in 1 × Tris Base SDS-page running buffer under reducing conditions at 150 V. Following electrophoretic separation, proteins were transferred to nitrocellulose membranes using the mini trans-blot electrophoretic transfer cell (Bio-Rad, Hercules, CA, USA) at 100 V for 90 min. Membranes were then blocked in 5% skim milk in PBST (0.05% Tween 20, phosphate-buffered saline) for 1 h and subsequently incubated with primary antibody overnight at 4 °C (Table 2). The membranes were then incubated with secondary antibody (HRP-conjugated) for 1 h (Table 2). SuperSignal Pico Chemiluminescent substrate (Thermo Fisher Scientific, Waltham, MA, USA) blotting detection reagents were used according to the manufacturer’s instructions. To detect faint signals, SuperSignal West Femto (Thermo Fisher Scientific, Waltham, MA, USA) was used according to the manufacturer’s instructions. Membranes were imaged in a Chemidoc touch detection unit (Bio-Rad, Hercules, CA, USA).

The membrane was then stripped with 0.5 M NaOH for 10 min and stained for *β-Actin* as a loading control for 1 h. The membrane was incubated with an HRP-conjugated secondary antibody for 1 h and imaged as previously described.

Quantitative densitometry of immunoblots was used to calculate changes in expression levels of the proteins of interest relative to the *β-actin* control. ImageJ (NIH, Bethesda, MD, USA) was used to calculate the intensity of individual bands, after which they were normalized to the loading control.

### 4.7. Immunocytochemistry and Microscopy

NRVMs plated on glass coverslips and transduced with rAAV for 3 weeks were fixed with 4% paraformaldehyde (PFA, Sigma-Aldrich, St. Louis, MO, USA) for 15 min, permeabilized with 0.3% Triton X-100 (Sigma-Aldrich, St. Louis, MO, USA) in PBS for 10 min and blocked with serum-free protein block (Dako, Copenhagen, Denmark) for 1 h. Samples were then incubated with primary antibody (Table 3) for 1 h at room temperature with mild shaking. Following 3 washes, the appropriate secondary antibody (Table 3) was applied for 1 h at room temperature in the dark. Samples were stained with DAPI (Sigma-Aldrich, St. Louis, MO, USA) before being mounted with glycerol-based mounting media onto glass slides. The slides were sealed and stored at 4 °C. Z-stack imaging was performed on an Olympus FV1000 confocal laser scanning microscope (Olympus, Tokyo, Japan) using a 60 × objective (Olympus 60 ×/1.35 NA Oil UPLFLN). All images were deconvolved with Huygens Professional software version 18.04 (Scientific Volume Imaging, Hilversum, The Netherlands, https://svi.nl/HomePage (accessed 27 July 2017)) using the CMLE algorithm, with SNR:30 and 40 iterations. Before being processed with ImageJ (NIH, Bethesda, MD, USA).

### 4.8. Flow Cytometry Analysis

NRVM cells for *GFP* analysis were harvested by trypsinization before being washed twice in PBS, pelleted and resuspended in PBS containing 1% FBS. Cells for *hTBX18* analysis were harvested by trypsinization before being washed, fixed in 1% PFA for 1 min and permeabilized in 0.05% Triton X-100 in PBS for 5 min. The cells were then blocked in PBS + 3% BSA for 30 min at room temperature before being subject to primary and secondary antibody incubations for 1 h each in the dark. The cells were then pelleted and resuspended in PBS containing 1% FBS. *GFP* and *hTBX18* fluorescence were assessed using a BD FACS Canto II (BD Biosciences, Franklin Lakes, NJ, USA) flow cytometer. Cell-size and singlet gates were set based on forward and side scatters, and the frequency of *GFP*+ and *hTBX18*+ cells was analyzed using FlowJo v10.4 (FlowJo LLC, Ashland, OR, USA).

### 4.9. Lentiviral Arclight Production, Titration and Transduction

Arclight, a voltage-sensitive-based fluorescent protein, was used to analyze action potentials generated by NRVMs [71,73]. Lentiviral Arclight was packaged in HEK293T cells by transfection using calcium phosphate precipitation with ppt.Arclight (Addgene, Watertown, MA, USA), pMDLgpRRE, pRSVREV, pMD2VSVG plasmids as previously described [82]. Lentivirus-containing supernatants were harvested at 48 and 72 h post-transfection, filtered through 0.45 µM syringe filters (Merck Millipore, Burlington, MA, USA), and concentrated using vivaspin 300,000 MWCO ultrafiltration tubes (Sartorius, Göttingen, Germany). Titer was assigned via functional transduction experiments in which dilutions of lentivirus stocks were used to transduce HEK293T cells, genomic DNA was extracted (Bioline, Memphis, TN, USA), and qPCR was performed to calculate the functional titer. The final titer of *LV.Arclight* was ~3 × 10^8^ transducing units/mL.

### 4.10. Arclight Confocal Line Scanning

NRVMs plated in glass bottom confocal dishes and co-transduced with rAAV6 and *LV.Arclight* for 3 weeks were analyzed for functional changes using a live cell imaging chamber on an Olympus FV1000 confocal laser scanning microscope. This involved repetitively scanning the same line 20,000 times on individual cells at a frequency of 850 Hz (~25 s) to measure changes in fluorescence intensity of Arclight over time [71,73]. To ensure consistency across treatment groups, cells were paced at a rate of 0.5 Hz using a custom-built cell culture pacing device.

Captured line scans were analyzed using a custom-written MATLAB program (MathWorks, Natick, MA, USA). This program corrected for baseline signal bleaching, filtered the signal to allow for better signal visualization and normalized, and inverted and zero-corrected it to allow signal parameters to be measured and compared across scans. APD50 was measured as the interval between the timing of 50% maximal upstroke amplitude and the time point of 50% of repolarization. Additionally, the peak-to-peak interval was used to confirm consistent beating frequencies.

### 4.11. Statistical Analysis

All data are presented as mean ± standard error of the mean. Differences between group means were compared using an unpaired *t*-test with Welch’s correction. Differences between several groups were compared using an ordinary one-way ANOVA. A value of *p* < 0.05 was considered statistically significant.

## 5. Conclusions

This study demonstrated the faithful recapitulation of the sophisticated pacemaker cell phenotype using an rAAV construct expressing *hTBX18* to reprogram ventricular myocytes, thus building on the previously employed stem cell and adenoviral-based approaches to biological pacing. This approach produced cells that not only converted on a molecular and physiological level but also underwent distinctive morphological and functional changes to become genuine SAN-like pacemaker cells. This novel method utilized a vector system that addresses some of the limitations of previous approaches and allows for new prospects for biological pacemakers to be developed in a clinically relevant gene transfer vector. Although limited by use in primary cells only, this method serves as a proof of principle in that NRVMs can be converted into genuine pacemaker cells using an rAAV vector. However, longer-term experiments in small and large animal models will be required to assess the efficacy, reliability and safety of this system before translation to patients who exhibit bradycardia.

## Figures and Tables

**Figure 1 ijms-23-09230-f001:**
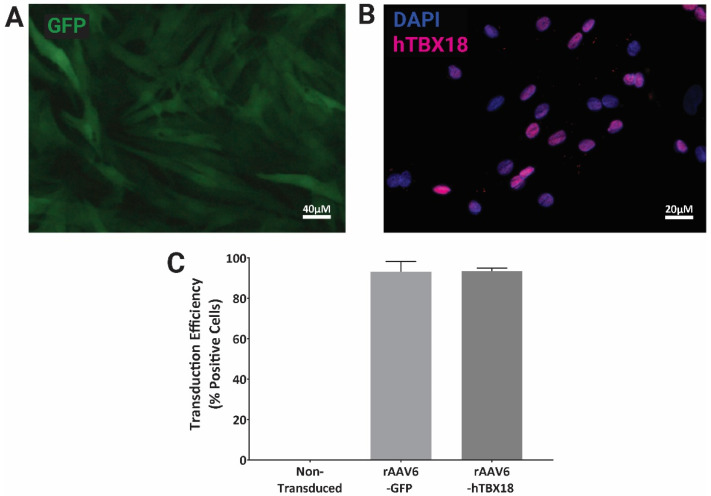
Efficient transduction of NRVMs using rAAV6 vectors. rAAV6-GFP and rAAV6-*hTBX18* transduction efficiency in NRVMs as visualized via microscopy and quantified via flow cytometry. (**A**) *GFP* fluorescence in NRVMs transduced with rAAV6-*GFP.* (**B**) NRVMs transduced with rAAV6-*hTBX18*, then co-stained with *hTBX18* antibody and DAPI. (**C**) Transduction efficiencies of NRVMs transduced with rAAV6-*GFP* and rAAV6-*hTBX18*, as quantified via flow cytometry. Data presented as mean ± SEM. n = 2 replicates.

**Figure 2 ijms-23-09230-f002:**
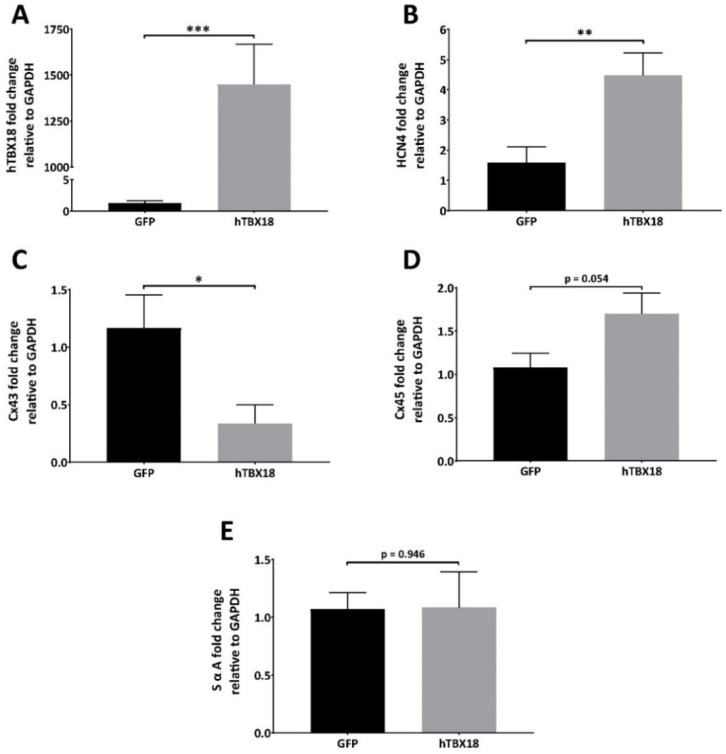
mRNA expression profile of NRVMs converted to pacemaker cells following overexpression of *hTBX18*. Quantitative analysis of changes in mRNA expression levels between rAAV6-GFP and rAAV6-*hTBX18*-transduced cells for the following genes. (**A**) *hTBX18*, (**B**) HCN4, (**C**) Cx43, (**D**) Cx45, (**E**) *SαA*. Data presented as mean ± SEM. *** *p*-value ≤ 0.001, ** *p*-value ≤ 0.01, * *p*-value ≤ 0.05.

**Figure 3 ijms-23-09230-f003:**
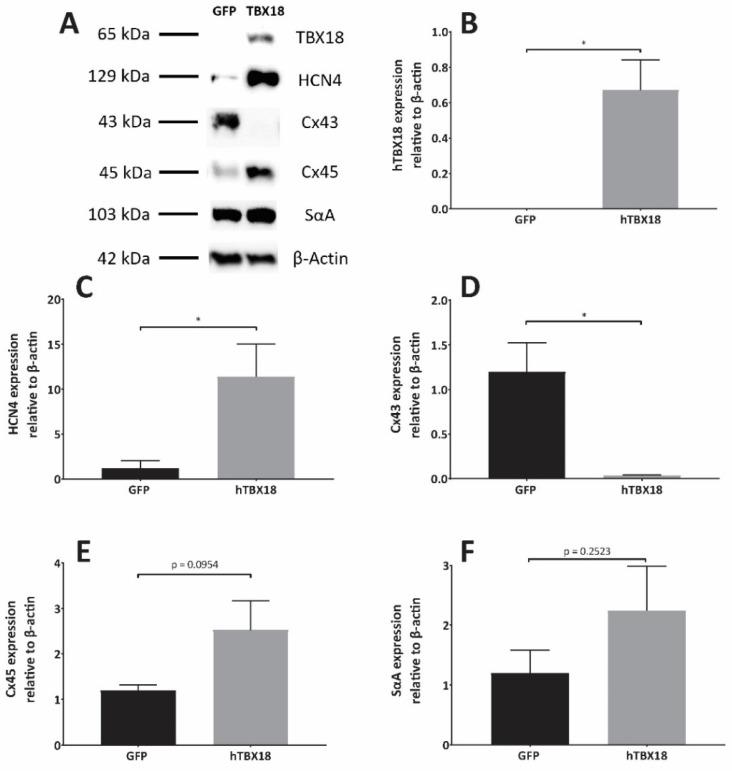
Protein expression profile of NRVMs converted to pacemaker cells following induced overexpression of *hTBX18*. Quantitative analysis of changes in protein expression levels between rAAV6-GFP- and rAAV6-*hTBX18*-transduced cells. (**A**) Representative figure showing protein bands generated by Western blot, their corresponding band sizes and the *β-actin* loading control (**B**) Quantification of *hTBX18* protein levels in cells transduced with rAAV6-*hTBX18* and rAAV6-GFP-transduced cells. (**C**) Quantification of HCN4 protein levels in cells transduced with rAAV6-*hTBX18* and rAAV6-GFP-transduced cells. (**D**) Quantification of Cx43 protein levels in cells transduced with rAAV6-*hTBX18* and rAAV6-GFP-transduced cells. (**E**) Quantification of Cx45 protein levels in cells transduced with rAAV6-*hTBX18* and rAAV6-GFP-transduced cells. (**F**) Quantification of *SαA* protein levels in cells transduced with rAAV6-*hTBX18* and rAAV6-GFP-transduced cells. Data presented as mean ± SEM. * *p*-value ≤ 0.05.

**Figure 4 ijms-23-09230-f004:**
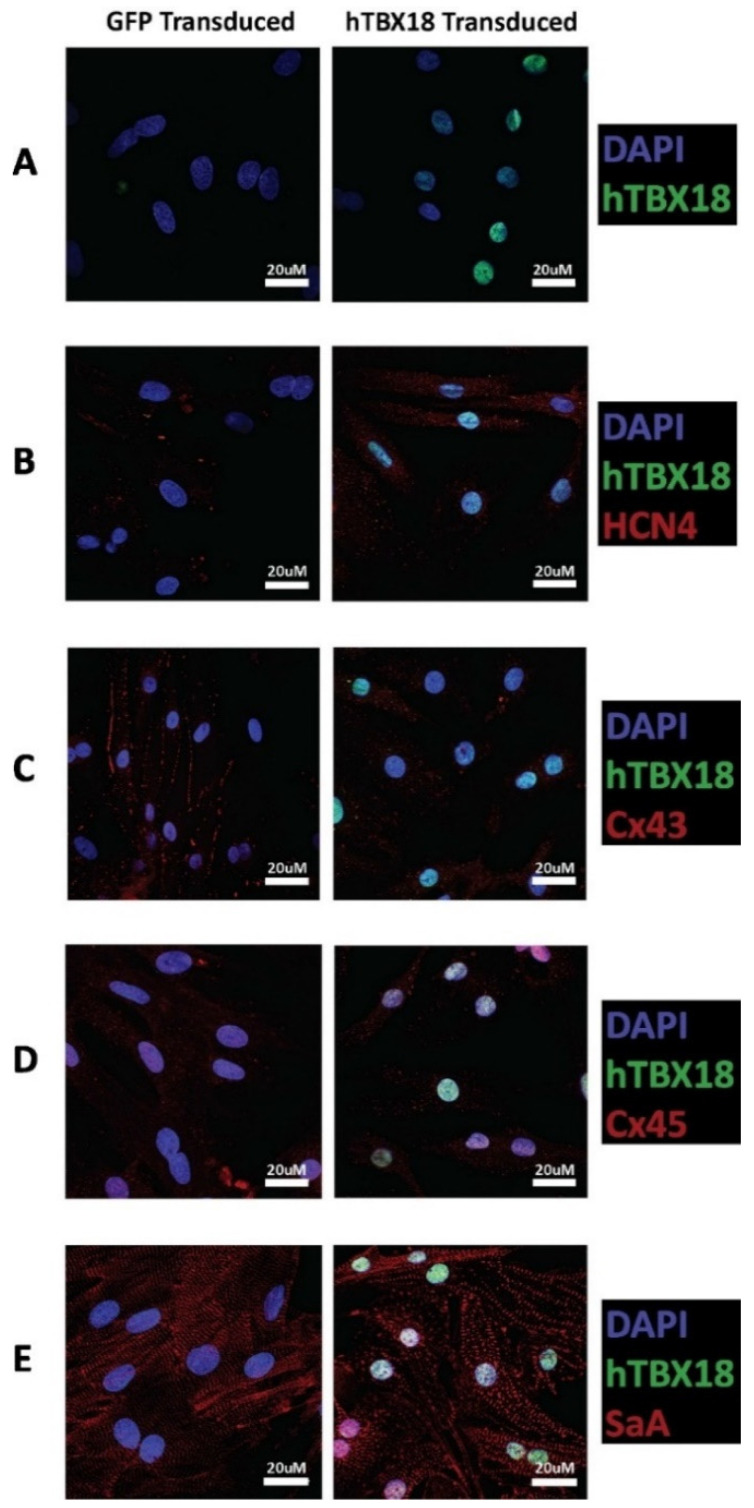
Characterization of expression levels of pacemaker markers in rAAV6-GFP and rAAV6-*hTBX18*-transduced cells by immunocytochemistry. (**A**) *hTBX18* (green) expression and (**B**) HCN4 (red) and *hTBX18* (green) expression. (**C**) Cx43 (red) and *hTBX18* (green) expression. (**D**) Cx45 (red) and *hTBX18* (green) expression. (**E**) *SαA* (red) and *hTBX18* (green) expression. All samples were also co-stained with DAPI to mark the nuclei (blue).

**Figure 5 ijms-23-09230-f005:**
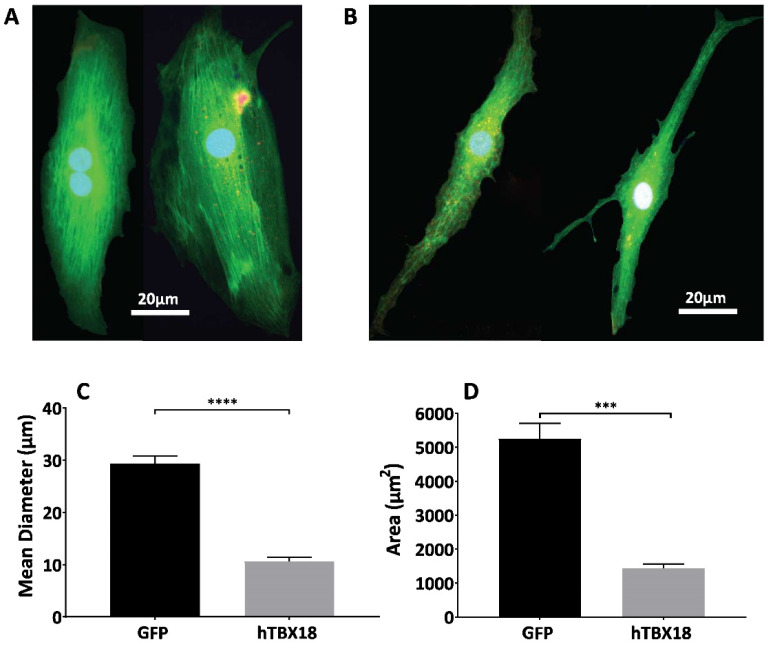
Morphological characterization of NRVMs converted to a pacemaker cell phenotype following induced overexpression of *hTBX18*. (**A**) Representative figures showing original NRVMs with their characteristic large, thick, block-like structure. (**B**) Representative figures showing rAAV6-*hTBX18*-transduced NRVMs which have become thin, spindle-like and tapering in structure after 3 weeks. (**C**) Quantification of the mean diameter of the transduced cells. (**D**) Quantification of the mean area of the transduced cells. Data presented as mean ± SEM. **** *p*-value ≤ 0.0001, *** *p*-value ≤ 0.001.

**Figure 6 ijms-23-09230-f006:**
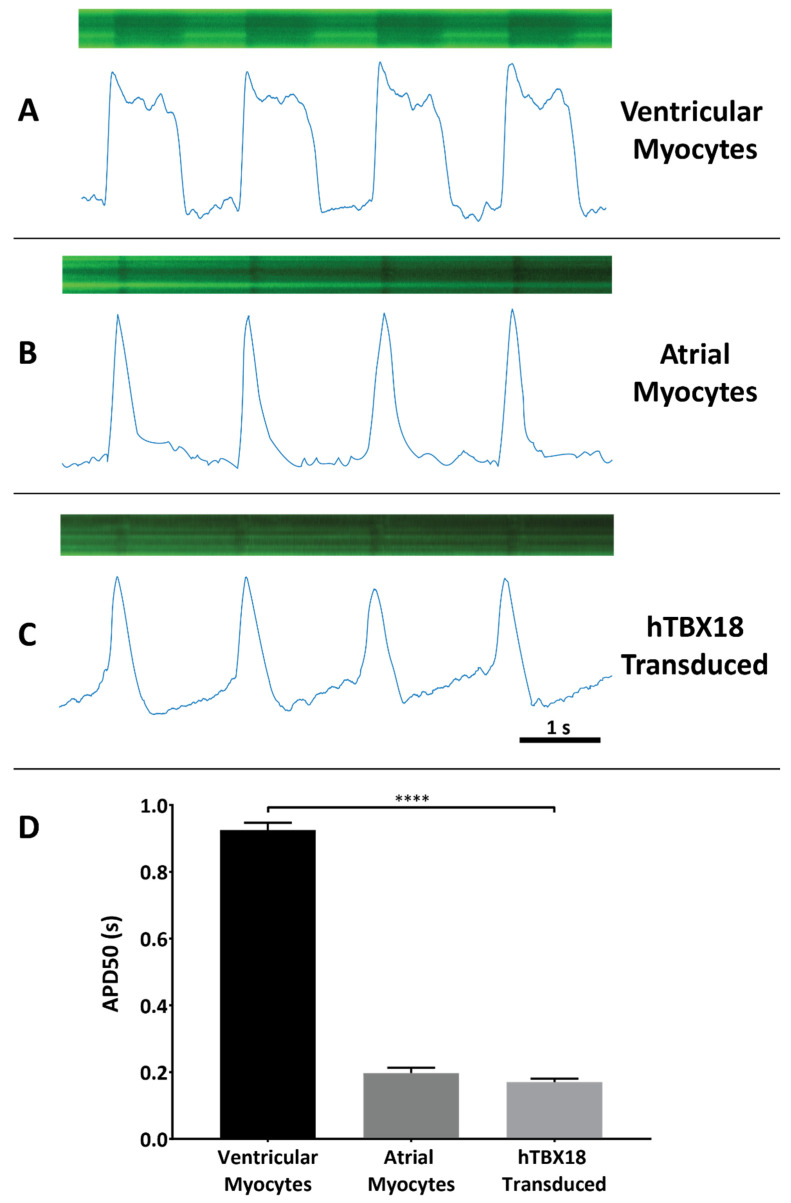
Pacemaker-like cell function was observed in *hTBX18*-transduced NRVMs. Action potentials recorded and analyzed from NRVMs and *hTBX18*-transduced cells using LV. Arclight co-transduction. Representative images of Arclight expression as generated by the confocal line scan protocol and the action potentials generated after analysis via MATLAB for (**A**) NRVM cells, (**B**) atrial myocyte cells (**C**) and rAAV6-*hTBX18*-transduced cells. (**D**) Quantification of the AP morphology using the APD50 measure. Data presented as mean ± SEM. **** *p*-value ≤ 0.0001.

**Figure 7 ijms-23-09230-f007:**
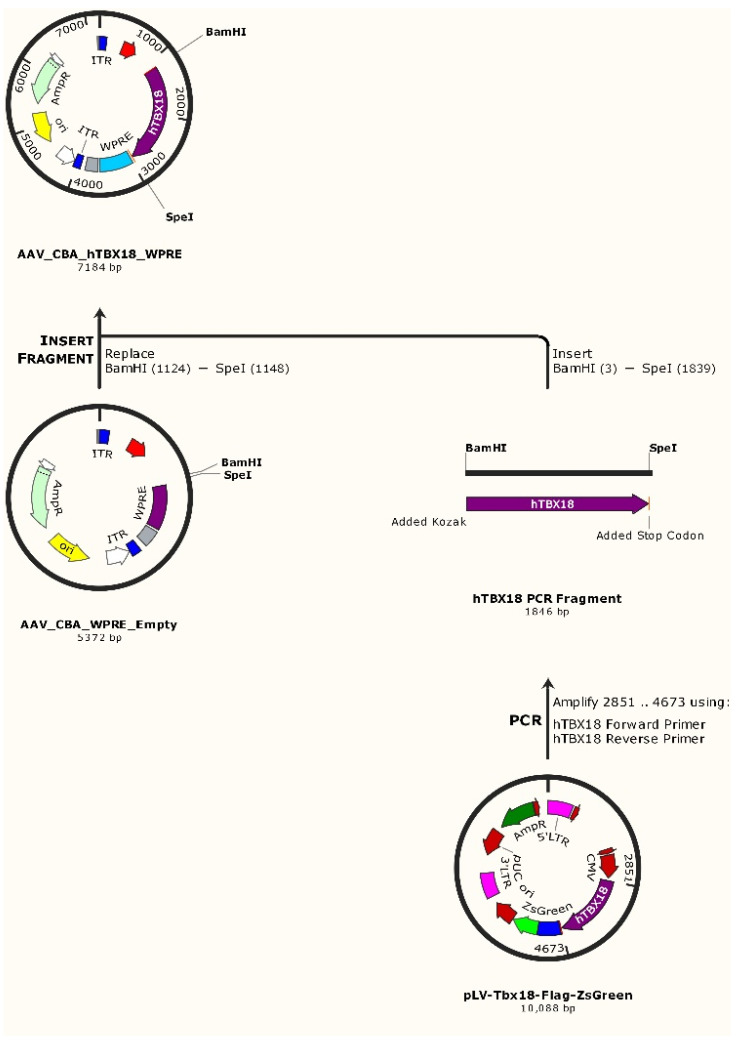
The cloning strategy followed to create the AAV_CBA_*hTBX18*_WPRE rAAV construct. This involved PCR amplifying the *hTBX18* gene fragment from a lentiviral backbone and sub-cloning it into the rAAV construct.

**Table 1 ijms-23-09230-t001:** Primers used for RT-PCR protocol to measure fold changes in synthesized cDNA.

Primer	Direction	Sequence
Human TBX18	Forward (5′-3′)	TTCTGGCGACCATCACTACG
Human TBX18	Reverse (5′-3′)	ACGCCATTCCCAGTACCTTG
Rat *HCN4*	Forward (5′-3′)	CGCATCCACGACTACTACGAAC
Rat *HCN4*	Reverse (5′-3′)	GGTCTGCATTGGCGAACAG
Rat *Cx43*	Forward (5′-3′)	AGCCTGAACTCTCATTTTTCCTT
Rat *Cx43*	Reverse (5′-3′)	CCATGTCTGGGCACCTCT
Rat *Cx45*	Forward (5′-3′)	TGCCTACAAGCAGAACAAAGC
Rat *Cx45*	Reverse (5′-3′)	TCCTCGTGGCTGCCATAC
Rat Actn2	Forward (5′-3′)	CTATTGGGGCTGAAGAAATCGTC
Rat Actn2	Reverse (5′-3′)	CTGAGATGTCCTGAATGGCG
Rat GAPDH	Forward (5′-3′)	GCATCACCCCATTTGATGTT
Rat GAPDH	Reverse (5′-3′)	TGGGAAGCTGGTCATCAAC

**Table 2 ijms-23-09230-t002:** Primary and secondary antibodies used in the immunoblotting protocol.

Primary Antibody	Source	Catalogue Number	Dilution
Goat anti-*TBX18*	Santa Cruz	sc-17869	1:200
Mouse anti-*HCN4*	Abcam	ab85023 (S114-10)	1:250
Rabbit anti-*Cx43*	Merck Millipore	Ab1727	1:500
Rabbit anti-*Cx45*	Koval	Donation	1:500
Mouse anti-*Sα**A*	Sigma	A7811	1:3000
Rabbit anti-*β-Actin*	Abcam	Ab8227	1:5000
**Secondary Antibody**	**Source**	**Catalogue Number**	**Dilution**
Anti-rabbit HRP	Thermofisher	31460	1:10,000
Anti-mouse HRP	Dako	P0447	1:10,000
Anti-goat HRP	Dako	P0449	1:10,000

**Table 3 ijms-23-09230-t003:** Primary and secondary antibodies used in the immunocytochemistry protocol.

Primary Antibody	Source	Catalogue Number	Dilution
Goat anti-*TBX18*	Santa Cruz	sc-17869	1:150
Mouse anti-*HCN4*	Abcam	ab85023 (S114-10)	1:150
Rabbit anti-*Cx43*	Merck Millipore	MAB3068	1:250
Rabbit anti-*Cx45*	Merck Millipore	MAB3100	1:150
Mouse anti-*SαA*	Sigma	A7811	1:1000
**Secondary Antibody**	**Source**	**Catalogue Number**	**Dilution**
Donkey anti-Goat Alexa fluor 594	Life Technologies	A-11058	1:1000
Donkey anti-Goat Alexa fluor 647	Invitrogen	A-21447	1:1000
Rabbit anti-Mouse Alexa fluor 594	Molecular Probes	A-11067	1:1000

## Data Availability

The data that supports the findings of this study are available from the corresponding author upon reasonable request.

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
