# Peer review of "Recombinant Adeno-Associated Viral Vector-Mediated Gene Transfer of hTBX18 Generates Pacemaker Cells from Ventricular Cardiomyocytes"

_ijms, 2022, doi:10.3390/ijms23169230_

Round 1
Reviewer 1 Report
Farraha report an in vitro study where they tested AAV6 based hTBX18 gene transfer in neonatal rat cardiomyocytes. They demonstrated that hTBX18 over-expression reprograms cells and provided convincing data that they show similarity to pacemaker cells. Since AAV based gene therapy can offer long-term gene expression, this approach might address previous limitations of gene therapy using adenovirus. I have only a couple of comments.
Figure 6 is nice showing the similarity between hTBX18 transduced cells and atrial cells. But other figures are only comparing the difference between GFP vs hTBX18 cells and lack atrial myocyte controls. Addition of atrial myocyte controls is suggested wherever possible.
There are some places with “Error! Reference source not found.”. Please fix.
Reviewer 2 Report
The paper by Farraha et al. describes the in vitro effects of reprogramming cardiomyocytes to pacemaker-like SAN cells by viral-mediated expression of the transcription factor hTBX18. The fascinating aspect of this study was that differentiated cells could be ‘directly’ reprogrammed to another differentiated cell-type without the necessity to de-differentiate the cardiomyocytes to pluripotent stem cells. Thus, the approach holds great promise to be incorporated in translational approaches to curing heart disease. In general, experiments were thoroughly conducted and included the proper controls. Nevertheless, a few minor issues should be addressed before deciding to accept the paper for publication.
l.71/72: ’…the triggering of the genetic…’
l.118: explain ‘NRVMs’
l.122: AAV transduction was performed for three weeks. Did the authors try shorter transduction times and what were the results?
l.122 & Fig.1; according to the images shown, it seems as if Fig.1B contains less cells than Fig.1A. Please comment and/or provide better images!
l.137ff: the authors should state (here) that expression levels of the different genes were analyzed by qPCR and GAPDH expression was used to normalize the values. Fig.2A, however, is misleading because the authors compare GFP expression to hTBX18 expression in transduced cardiomyocytes. According to the experimental design, samples were either transduced by the one or the other AAV-construct. Therefore, the hTBX18 transcription factor can only be detected in the AAV-hTBX18 transduced sample but not in AAV-GFP transduced cells and vice versa. This should be corrected accordingly! In addition, stating that values differ ‘significantly’ needs to be supported by SD or SEM values (throughout the ms)!
l.166 & Fig.3; (A) one hardly recognizes a signal in hTBX18 transduced cells for Cx43. This should be explained because in the densitometric analysis it seems that protein was present! As already mentioned for Fig.2, the authors should omit comparing hTBX18 and GFP levels (3B)! Furthermore, it would be interesting to know whether, e.g., HCN2 channel subunit expression was affected by the treatments because HCN2 gene expression typically outnumbers that of the other isoforms.
l.183ff & Fig4; the immunological data should be described more patiently. Especially expression of the HCN4 subunit is hardly detected in the plasma membrane rather than in intracellular structures. This might be due to the antibody chosen. In this respect, have the authors purchased and tested independent antibodies? To support localization of proteins in the cell membrane, the authors should provide better images and also samples counterstained with membrane markers.
l.314ff: wasn’t it surprising that hTBX18 acted as an expression activator (e.g. at the hcn4 promoter) and simultaneously as an expression inhibitor at the Cx43 promoter? Shouldn’t additional factors be considered in mediating these different effects? This issue should be included in the discussion.
l.370/372: exchange ‘digest’ for ‘restriction’
l.374: it would be interesting to know whether the authors tested constructs with viral promoters, e.g. CMV, instead of CBA to drive transgene expression.
l.399: give details of the cesium chloride gradient
l.402: how was viral DNA prepared for qPCR analyses?
l.470: ‘…4x Laemmli…’
